# Measurements of Spatial Angles Using Diamond Nitrogen–Vacancy Center Optical Detection Magnetic Resonance

**DOI:** 10.3390/s24082613

**Published:** 2024-04-19

**Authors:** Zhenrong Shi, Haodong Jin, Hao Zhang, Zhonghao Li, Huanfei Wen, Hao Guo, Zongmin Ma, Jun Tang, Jun Liu

**Affiliations:** 1School of Instrument and Electronics, North University of China, Taiyuan 030051, China; zhenrongshi@yeah.net (Z.S.); guohao@nuc.edu.cn (H.G.); 2School of Semiconductors and Physics, North University of China, Taiyuan 030051, China; 18834169805@163.com (H.J.);

**Keywords:** optically detected magnetic resonance, measurements of spatial angles, magnetic microscope, diamond nitrogen–vacancy center

## Abstract

This article introduces a spatial angle measuring device based on ensemble diamond nitrogen–vacancy (NV) center optical detection magnetic resonance (ODMR). This device realizes solid-state all-optical wide-field vector magnetic field measurements for solving the angles of magnetic components in space. The system uses diamond NV center magnetic microscope imaging to obtain magnetic vector distribution and calculates the spatial angles of magnetic components based on the magnetic vector distribution. Utilizing magnetism for angle measuring enables non-contact measuring, reduces the impact on the object being measured, and ensures measurement precision and accuracy. Finally, the accuracy of the system is verified by comparing the measurement results with the set values of the angle displacement platform. The results show that the measurement error of the yaw angle of the system is 1°, and the pitch angle and roll angle are 1.5°. The experimental results are in good agreement with the expected results.

## 1. Introduction

The application of spatial angle measurement is very extensive, involving multiple fields such as materials science, biomedicine, control systems, aerospace, and aviation, and it is of great significance for research and practical applications [1,2,3,4]. Angle measurement is mainly divided into two types: contact and non-contact. By utilizing contact measurement, it is possible to attain a higher angular resolution. However, with increased measurement time, wear occurs, necessitating frequent maintenance and replacement, thus increasing measurement costs and usage difficulties [5,6]. On the other hand, non-contact angle measurement overcomes these disadvantages. Common non-contact angle measurement methods include optoelectronic angle measurement and magnetic angle measurement. Although optoelectronic angle measurement provides high accuracy, it has strict requirements for the testing environment and may not be suitable for complex testing environments. Magnetic angle measurement offers advantages such as high precision, high resolution, ease of miniaturization, and suitability for harsh environments, making it a widely recognized angle measurement method [7]. However, conventional magnetic angle measurement methods often require multiple vector magnetic sensors to be assembled around a magnetic ring, leading to systematic errors in angle measurement due to unavoidable assembly errors and zero drift [8,9,10,11]. In recent years, the magnetic field imaging technology based on the diamond nitrogen–vacancy (NV) center has attracted widespread attention from researchers due to its high magnetic field detection sensitivity at room temperature and stable crystal structure. Particularly, the ensemble of the diamond NV center, formed by many NV centers integrated into a single diamond crystal with inherent crystallographic axes, possesses built-in vector magnetic field sensing characteristics, effectively eliminating errors introduced by sensor misalignment with coordinate axes. Wide-field magnetic vector imaging based on an ensemble diamond NV center can simultaneously obtain a large-scale two-dimensional vector magnetic field array, providing rich magnetic field data sets for angle detection [12,13,14]. Diamond NV centers are expected to offer new methods for magnetic angle measurement.

Since the combination of the diamond NV center and magnetic resonance technology in 2008 [15], it has attracted increasing attention from researchers due to its excellent spin and optical properties. The diamond NV center is a nanomaterial with combined optical and magnetic resonance capabilities, holding broad application potential in quantum information and biomedical fields [16,17,18]. The optical detection magnetic resonance (ODMR) technology primarily exploits the characteristics of a diamond NV center by laser exciting the NV center to specific energy levels, manipulating the spin state of the NV center through microwave radiation, and finally obtaining information on the magnetic field intensity of the sample by detecting changes in its fluorescence intensity [19,20]. The magnetic field is a fundamental physical property of various substances. Magnetic components or electronic devices that generate electromagnetic effects when powered (such as chips or PCBs) will radiate a magnetic field in their surroundings. This magnetic field carries a large amount of physical information [21,22], and spatial angle information can be deduced by measuring the magnetic field. This article introduces a spatial angle detection device based on NV center photodetection of magnetic resonance, utilizing the high-precision magnetic sensing capability of the diamond NV center to achieve magnetic angle measurement.

This article is based on the study of the magnetic field imaging method of the ensemble diamond NV center. By utilizing continuous wave ODMR magnetometry, we realized the detection of the spatial angle of the magnetic component. First, we investigated the principle of diamond NV center magnetic field detection. Then, we analyzed the theoretical basis of determining the spatial angle of the test component based on the distribution information of the magnetic vector field and investigated the method for evaluating the measurement accuracy of the system. Finally, based on our self-built ODMR imaging system, utilizing the magnetic field information of the target components captured by the camera on a large-ensemble diamond, the spatial angles of the test sample were obtained according to the magnetic field distribution. To verify the accuracy of the system, we conducted angle tests on the target chip, resulting in a pitch angle error of 1° and roll and pitch angle errors of 1.5° each. The achieved angle resolution similar to the angle sensor recently reported [7] provided a new method for angle detection.

## 2. Experimental Principles

### 2.1. Magnetic Field Detection Based on ODMR

As shown in Figure 1a, the NV center is composed of a nitrogen atom (N) and a vacancy (V) replacing a carbon (C) atom [23]. It can be divided into NV^0^ and NV^−^ states based on its fluorescence charge state. Due to the sensitivity of the NV^−^ center to magnetic fields [24], this study focuses on the NV^−^ center ensemble. For ease of expression, we will use the term NV center to refer to NV^−^ in the following text. The magnetic field detection of the diamond NV center is a magnetic resonance technique based on ODMR. The main principle is as follows: firstly, polarize the NV center to the initial state with a 532 nm laser; then, manipulate the quantum state with microwaves and determine the current quantum state of the NV center by counting fluorescent photons while polarizing them with the laser; finally, achieve high-sensitivity magnetic field measurement by utilizing the sensitivity of the spin system to magnetic fields. The Hamiltonian of the ground-state electron spin of the NV center [16,17] can be represented as:(1)$$H=DS_{z}^{2}+g\mu_{B}B_{z}S_{z}+E_{1}\left(S_{x}^{2}-S_{y}^{2}\right)$$
where the first term is the zero-field splitting of the electron spin in the NV axis direction (Z direction), where *D* = 2.87 GHz is the zero-field splitting constant. During NV center magnetic detection, it is necessary to apply a bias magnetic field *B*_bias_, where the bias magnetic field magnitude needs to satisfy *B*_bias_ ≫ *hD*/*g_μ_B* in order to use the bias magnetic field to counteract the influence of local stress (*E*) on the diamond, which is *E* = 0. *S_X_*, *S_Y_*, and *S_Z_* represent the spin angular momentum operators of the NV electron spin, with a spin quantum number of 1; the third term is the Zeeman term of the static magnetic field, which can be used for external magnetic field detection. *S* represents a vector with components *S_X_*, *S_Y_*, and *S_Z_*, *g* is the Landé *g*-factor, and *µ_B_* is the Bohr magneton. The resonant frequency of ODMR in the NV axis direction can be derived to be [25]
(2)$$\Delta v=v^{+}-v^{-}=\frac{g\mu_{B}}{h}B$$
where *h* = 6.63 × 10^−34^ J·s is the Planck constant; *ν*^+^ and *ν*^−^ are the positive and negative electron spin resonance.

The resonance frequency is linearly related to the projection of the magnetic field on the NV axis. Therefore, by measuring the ODMR characteristics of the ensemble NV center, the distribution of the magnetic field on the four crystal axes can be obtained. The projections of the magnetic field on the four NV axes obtained through ODMR characteristics can be used to detect the vector magnetic field [26,27]. As shown in Figure 1b, the ground-state electron energy level ^3^*A*_2_ exhibits a zero-field splitting (ZFS) of *D* = 2.87 GHz between the spin states with *ms =* 0 and *ms* = ±1. In the presence of an external magnetic field, the electron spin of the NV center will undergo Zeeman splitting, with the projection along the diamond NV axis linearly dependent on the magnetic field. Through ODMR spectra (as shown in Figure 1c), we can obtain four pairs of resonant frequencies, thus determining the magnetic field distribution along the four NV axes. Therefore, the projection magnetic fields *B*_1_, *B*_2_, *B*_3_, and *B*_4_ of the external magnetic field *B* will exist on the four NV center axes.
(3)$$\left\{ \begin{array}{l} B_{1}=\left({v_{1}}^{+}-{v_{1}}^{-}\right)\cdot \frac{h}{2g\mu_{B}}\\ B_{2}=\left({v_{2}}^{+}-{v_{2}}^{-}\right)\cdot \frac{h}{2g\mu_{B}}\\ B_{3}=\left({v_{3}}^{+}-{v_{3}}^{-}\right)\cdot \frac{h}{2g\mu_{B}}\\ B_{4}=\left({v_{4}}^{+}-{v_{4}}^{-}\right)\cdot \frac{h}{2g\mu_{B}} \end{array}\right.$$

The symmetry axes of the NV center and the directions of the three-axis Cartesian coordinate frame defined in the laboratory are shown in Figure 1d. Taking the [100] crystal orientation diamond we use as an example here, the normal to the sensing surface is aligned with the [100] crystal orientation, with the sensing surface located at the top of the diamond. The unit vectors *μ*_1_, *μ*_2_, *μ*_3_, and *μ*_4_ of the four NV center symmetry axes are defined along the crystal lattice vectors of the diamond. For ease of calculation, these unit vectors are all taken with respect to the positive component of the *Z*-axis. Therefore, the four-unit vectors *μ*_1_, *μ*_2_, *μ*_3_, and *μ*_4_ are as follows:(4)$$\left\{ \begin{array}{l} \vec{u_{1}}=\left(-\sqrt{\frac{2}{3}},0,\sqrt{\frac{1}{3}}\right)=\frac{1}{\sqrt{3}}\left(-\sqrt{2},0,1\right)\\ \vec{u_{2}}=\left(\sqrt{\frac{2}{3}},0,\sqrt{\frac{1}{3}}\right)=\frac{1}{\sqrt{3}}\left(\sqrt{2},0,1\right)\\ \vec{u_{3}}=\left(0,\sqrt{\frac{2}{3}},\sqrt{\frac{1}{3}}\right)=\frac{1}{\sqrt{3}}\left(0,\sqrt{2},1\right)\\ \vec{u_{4}}=\left(0,-\sqrt{\frac{2}{3}},\sqrt{\frac{1}{3}}\right)=\frac{1}{\sqrt{3}}\left(0,-\sqrt{2},1\right) \end{array}\right.$$

By combining Equations (3) and (4), we can obtain
(5)$$\left\{ \begin{array}{l} \sqrt{3}B_{1}=-\sqrt{2}B_{x}+B_{z}\\ \sqrt{3}B_{2}=\sqrt{2}B_{x}+B_{z}\\ \sqrt{3}B_{3}=\sqrt{2}B_{y}+B_{z}\\ \sqrt{3}B_{4}=-\sqrt{2}B_{y}+B_{z} \end{array}\right.$$

As shown in Equation (5)
(6)$$\left\{ \begin{array}{l} B_{x}=\frac{\sqrt{6}}{4}\left(B_{2}-B_{1}\right)\\ B_{y}=\frac{\sqrt{6}}{4}\left(B_{3}-B_{4}\right)\\ B_{z}=\frac{\sqrt{3}}{4}\left(B_{1}+B_{2}+B_{3}+B_{4}\right) \end{array}\right.$$

The magnitude of magnetic field *B* can be expressed as follows:(7)$$B=\sqrt{{B_{X}}^{2}+{B_{Y}}^{2}+{B_{Z}}^{2}}$$

### 2.2. Magnetic Field Angle Detection Method

Figure 2a illustrates a schematic diagram of the method based on magnetic field positioning calculation [28]. This method considers an infinite straight conductor consisting of *N* identical current elements closely connected along the axis. Each current element can be simply regarded as a point neglecting its outer dimensions. Since the magnetic field should have cylindrical symmetry, it is possible to choose any point (0, *N*) on the conductor to study the magnetic field produced by the current element at that point on the diamond sensor surface at a distance of *R*.
(8)$$B=\int_{\theta =0}^{\theta =\pi }\frac{kIdl\sin\theta }{r^{2}}=kI\int_{0}^{\pi }\sin\theta \cdot \frac{\sin^{2}\theta }{R^{2}}\cdot \frac{Rd\theta }{\sin^{2}\theta }=\frac{kI}{R}{\left[-\cos\theta \right]}_{0}^{\pi }=\frac{2kI}{R}$$

Here, *k* is the proportionality coefficient, and the magnetic field *B* is directly proportional to the current *I* and inversely proportional to the distance *R*.

Since the magnetic field intensity produced by the current element is the same on the circle of its rotation axis, as shown in Figure 2a,b, when the diamond sensing surface is parallel to the straight wire, the magnetic field intensity detected at *R* is also the same. In Figure 2b, when an infinitely long straight current makes an angle *θ* with the diamond sensing surface, the detected magnetic field intensity increases with the distance of the current element from the NV center. The angular space between the magnetic field calculated for the nth current element is *θ_n_*:(9)$$\theta_{n}=\left(\frac{2kI}{B_{n}}-\frac{2kI}{B_{n+1}}\right)/d\;n=\left(1,2,3,\ldots ,N\right)$$

Here, *d* represents the distance between adjacent current elements. To improve accuracy, it is necessary to calculate the angles between all current elements and take the average, denoted as the final determined angle *θ*.
(10)$$\theta =\frac{1}{N}\sum_{i=1}^{N}\theta_{i}\;n=\left(1,2,3,\ldots ,N\right)$$

From the above calculation process, calculating the pitch *φ* and roll angle *β* requires extracting *B*_1_–*B_N_* from the magnetic field map. As shown in Figure 2b, we can visually assess the approximate positions of *B*_1_–*B_N_* on the magnetic field map. However, for a more accurate estimation of the magnetic field, we need to use a more precise method to locate *B*_1_–*B_N_*. Here, we take the positioning of a single wire as an example. This method involves grouping the magnetic field map by rows and columns, extracting the magnetic field strength of each pixel, and plotting a graph of the position against the magnetic field strength with pixel position as the horizontal axis and magnetic field strength as the vertical axis. Figure 3 shows a flowchart. After Lorentz fitting, the center of the full width at half maximum of each spectral line is determined for pixel positioning, and the extracted pixel point is denoted as (*a*, *b*). The actual coordinates of this point in the coordinate system are (*i*, *j*).
(11)(i,j)=(d×a,d×b)

Here, *d* represents the pixel size. This method can also determine the angle *α* between the wire and the *X*-axis in the laboratory Cartesian coordinate system, used to position the wire’s yaw angle in the plane.
(12)$$\alpha_{n}=\arctan\left(\frac{j_{n}-j_{n+1}}{i_{n}-i_{n+1}}\right)\;n=\left(1,2,3,\ldots ,N\right)$$

Here, *α_n_* represents the angle between the *n*–th current element and the *X*-axis, (*i_n_*, *j_n_*) is the position of the nth current element on the magnetic field diagram, (*i_n+_*_1_, *j_n+_*_1_) is the position of the (*n* + 1) th current element on the magnetic field diagram, and *α* is the final determined angle with the *X*-axis.
(13)$$\alpha =\frac{1}{N}\sum_{i=1}^{N}\alpha_{i}\;n=\left(1,2,3,\ldots ,N\right)$$

## 3. Experimental Apparatus

The core of the device is a 5 × 5 × 0.5 mm^3^ (100) oriented CVD diamond plate, with an estimated concentration of the NV center of about 1.5 ppm. As shown in Figure 4, the imaging system consists of six subsystems: the laser excitation system, microwave scanning system, fluorescence collection system, bias magnetic field system, solution system, and control system. The laser polarization system is primarily designed to provide a high-power, high-stability, and low-noise homogeneous laser for an ensemble of the diamond NV center. The laser excitation system is mainly composed of an MGL-FN-532 laser (Changchun Xingong Optoelectronics Technology Co., Ltd., Changchun, China). The main function of the microwave system is to provide a homogeneous microwave field for manipulating the electron spin of the diamond NV center. The microwave scanning system consists of a microwave source and an antenna, utilizing the high uniform microwave radiation characteristics we have developed to input a 94% resonant microwave field with a power of 30 dBm to the diamond NV center [29,30]. The fluorescence collection system mainly consists of an objective lens, a CCD camera, and a computer, transforming fluorescence signals into ODMR signals and calculating magnetic field and temperature information. When using the diamond NV center for magnetic measurements, it is generally necessary to separate the degenerate ±1 resonance peaks using a bias magnetic field. An experimental magnetic field adjustment device designed in the laboratory can apply over 95% uniform magnetic field space in the central square region of the diamond. In the experiment, by applying a bias magnetic field *B*_0_, with a magnitude of approximately 2.84 mT (*B*_0*X*_ = 1.23, *B*_0*Y*_ = 2.56, and *B*_0*Z*_ = 0.055 mT), four pairs of ODMR resonance peaks are obtained, enabling vector magnetic field detection. The control system mainly consists of an arbitrary signal generator, providing TTL pulse signals to control the coordinated operation of the other four systems, enabling pulse ODMR data acquisition.

The diamond NV center is imaged on the focal plane of the CCD, with a resolution of 1920 × 1080 pixels, and the area of a single pixel is 5 μm^2^. The CCD exposure frequency is 200 frames per second (FPS). The system uses a 10 × magnification, NA = 0.25 microscope objective, with an imaging field size of 960 × 540 μm^2^. The control system synchronizes the camera with the microwave source. The microwave source trigger mode is an external trigger, the microwave source output power is 0 dBm, and the power amplifier’s gain setting is +25 dBm, scanning the microwave source from 2.5 to 3.1 GHz, with a step frequency of 0.3 MHz. The imaging speed is 10 s based on the camera’s frame rate setting. We obtain a set of images, each representing a microwave frequency. The computer processes and analyzes many grayscale images transmitted by the CCD. The grayscale value of each pixel can be used to plot an ODMR, and the peak position is determined using Lorentzian fitting. The peak value is then used to estimate the magnetic field and ultimately generate a magnetic field map.

The NV center magnetic field imaging includes the following steps: First, the initialization of the diamond NV center is achieved by using a laser focused on the diamond. Then, through the scanning of the microwave frequency, the level inversion of the *ms* = ±1 state and *ms* = 0 state is achieved under the action of resonant microwaves. Finally, ODMR signals are collected using a camera. During this process, it is necessary to ensure the synchronization of microwave scanning frequency points with camera image acquisition, so that each image captured by the camera corresponds to the NV center fluorescence image at a specific microwave frequency point. By correlating the fluorescence intensity of all images with the scanning microwave frequency points, a complete ODMR spectrum can be obtained. Viewing the entire imaging area of the camera as an array of photodetectors, each pixel on the camera can be considered as a miniature photodetector.

After the camera finishes data collection, it is necessary to extract and process the ODMR fluorescence data and use the Lorentz fitting formula to obtain the fitted ODMR spectrum.
(14)$$f=y_{0}+\frac{2}{\pi }\sum_{i=1}^{n}\left(\frac{A\Gamma }{4{\left(x-x_{n}\right)}^{2}+\Gamma^{2}}\right)$$
where *y*_0_ is the fluorescence intensity value of the non-resonant segment of the ODMR spectrum, Γ is the line width, and *x_n_* is the microwave frequency of each resonance peak in the ODMR spectrum.

The sensitivity *η* of magnetic imaging detection based on the diamond NV center can be defined as the minimum detectable magnetic field when the signal-to-noise ratio (SNR) is 1 [31]. When fitting the ODMR spectrum with the Lorentz formula, the most sensitive magnetic detection can be obtained through the following formula [32]:(15)$$\eta =A_{\eta }\frac{h}{g_{e}\mu_{B}}\frac{\Gamma }{C\sqrt{I_{0}}}$$

Here, *A_η_* represents the line width correction factor, Γ denotes the line width, *C* stands for the contrast, *I*_0_ is the photon counting rate of the camera, and *h*/(*g_e_μ_B_*) ≈ 36 nT/KHz. Here, *A_η_* = 0.769, Γ is approximately 10 MHz, *C* is about 2%, and upon calculation, *η* is approximately 0.26 μT/Hz^1/2^.

## 4. Discussion

A 532 nm laser output from a laser is parallelly incident on the entrance pupil of the microscope by adjusting it, and the beam fills the entrance pupil. The microscope objective lens selected here has a magnification of 10× and NA = 0.3. The microscope lens focuses the laser on the diamond sensitive layer.

To evaluate the ability of angle detection, we tested the angles of the chip. Using a UV curable adhesive, the chip was fixed on a six-degree-of-freedom displacement stage. The displacement stage can precisely control the motion of three postures of yaw, pitch, and roll, as well as six degrees of freedom of *x*, *y*, and *z* translation, precisely controlling the distance and angle between the sample and diamond. A 20 Ω resistor was connected to the chip, and a current of 0.5 A flowed through the chip to generate an electromagnetic field when imaging was powered on. Initially, testing was conducted on the yaw angle by using the method of controlling variables, with both pitch and roll angles set to 0° and the distance between the sample and diamond set to 2 mm. To increase reliability, each set of data was tested 40 times and averaged. The magnetic field imaging results are shown in Figure 5a–e. The chip as shown in Figure 5f. The experimental results are as shown in the figures. As shown in Figure 6A, the experimental results indicate a maximum angular error of 1.1° and an average error of 0.95°. No significant trends were observed with the changes in rotation angle.

Then, we assessed the spatial pitch motion. At this time, the rotation table was at 0°, and the wire axis was parallel to the *Y*-axis of the diamond plane. The magnetic field imaging results are shown in Figure 7a–e. The experimental results shown in Figure 6A indicate that the maximum flipping angle error is 1.8°, with an average error of 1.28°. As the space flipping angle increases, the error gradually increases, which is caused by the increase in magnetic measurement error due to the increase in the distance between the detection object and the diamond.

Finally, under preset conditions, a comprehensive evaluation was conducted on the yaw angle *α*, pitch angle *φ*, and roll angle *β*. The magnetic field imaging results are shown in Figure 8a–j. The experimental results are shown in Figure 6B, indicating that the average error of yaw angle *α* is 1.1°. The average error of pitch angle *φ* is 1.33°, and the average error of roll angle *β* is 1.41°. The good consistency between the results of the two separate tests further confirms the stability of the coordinate testing being the same.

The impact of the spacing between the test sample and diamond on the angular resolution was tested and analyzed. We analyzed 10 different spacing groups, with specific preset spacing values and angular errors detailed in Table 1. As shown in Figure 9, through the analysis of the test results on spacing and angular resolution, it was found that as the spacing increases, the angular error also gradually increases, primarily influenced by the sensitivity of the magnetic field.

## 5. Conclusions

This paper presents a novel magnetic positioning system based on the high-sensitivity magnetic field detection of the diamond NV center, enabling the detection of plane and spatial angles by relating the spatial distance between magnetic fields and electromagnetic radiation. Using a chip as the test device, we achieved measurements of the plane yaw angle and spatial pitch angle. The results show that the measurement error of the yaw angle of the system is 1°, and the pitch angle and roll angle are 1.5°. The experimental results of the measurement system were compared with the expected ones, showing good consistency in both cases. Therefore, our positioning method is well suited for the spatial angle detection of magnetic components that are encapsulated or concealed. Furthermore, by combining with the scanning confocal system, it is expected to achieve wide-angle detection. Through enhancing control strategies, measurement accuracy can be further improved, providing new measurement methods for various application scenarios.

## Figures and Tables

**Figure 1 sensors-24-02613-f001:**
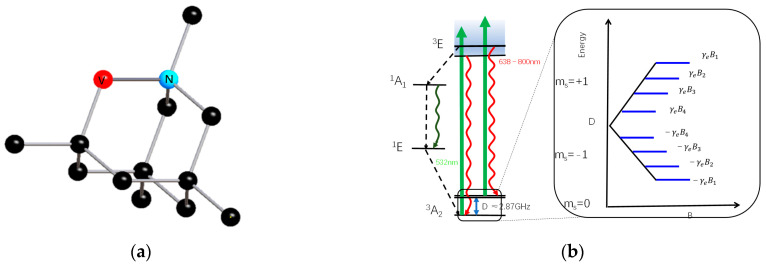
(**a**) Energy level diagram for the diamond NV center in the diamond. (**b**) The NV center in the diamond. (**c**) Optically detected magnetic resonance spectrum. (**d**) The NV axis and magnetic field *B* in the three-axis coordinate system.

**Figure 2 sensors-24-02613-f002:**
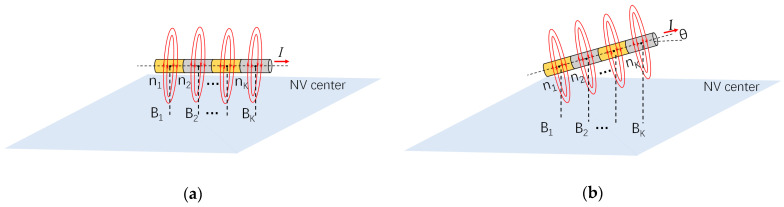
(**a**) Angle measurement based on the magnetic field when the wire is parallel to the diamond; (**b**) angle measurement based on the magnetic field when there is an angle *θ* between the wire and diamond.

**Figure 3 sensors-24-02613-f003:**
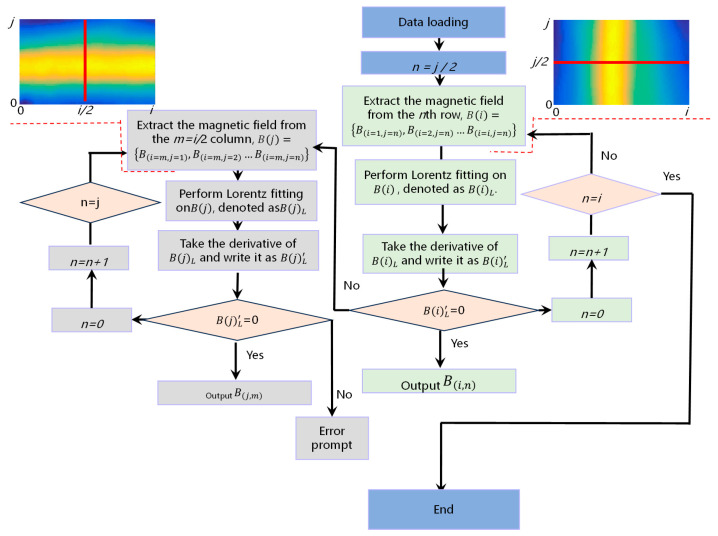
A flowchart for extracting the positions of corresponding rows and lines in a magnetic field map.

**Figure 4 sensors-24-02613-f004:**
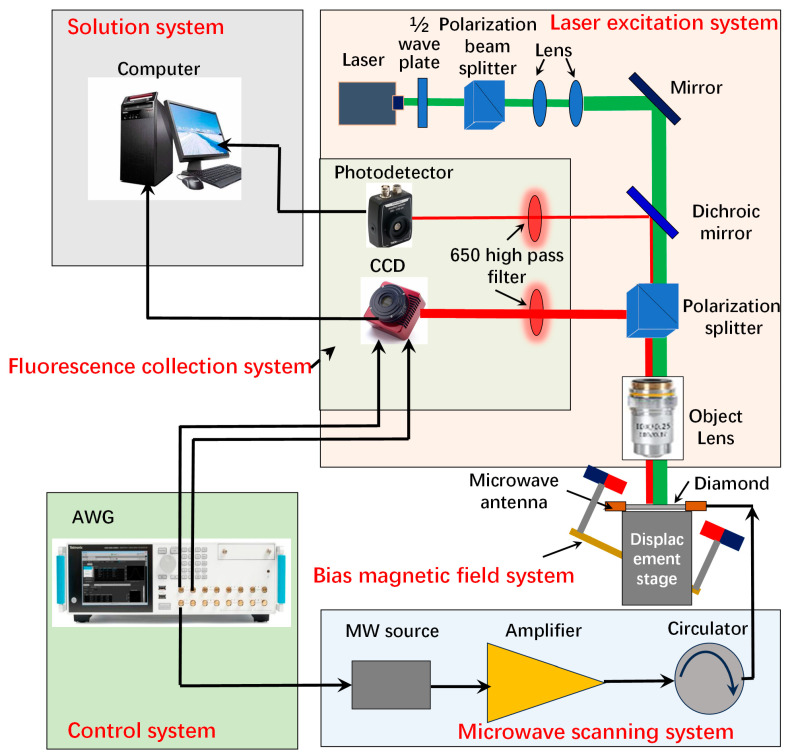
The schematic diagram of the experimental device.

**Figure 5 sensors-24-02613-f005:**
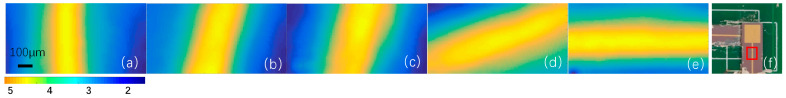
Test results for yaw angle α. (**a**–**e**) The magnetic field imaging results at yaw angles of 90°, 60°, 45°, 30°, and 0°. (**f**) Physical image of temperature control chip.

**Figure 6 sensors-24-02613-f006:**
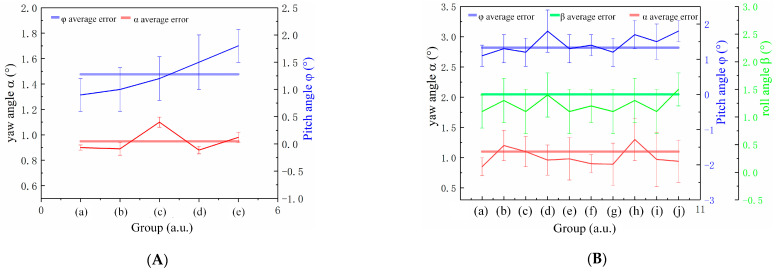
(**A**) Test results for yaw angle α and pitch angle φ of control variable method, (a)–(e) represent the corresponding 5 sets of test results in Figure 5 and Figure 7. (**B**) Test results for yaw angle *α*, pitch angle *φ*, and roll angle *β* of control variable method, (a)–(j) represent the corresponding 10 sets of test results in Figure 8.

**Figure 7 sensors-24-02613-f007:**
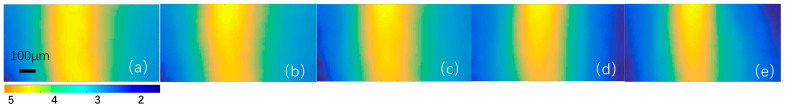
Test results for pitch angle φ.(**a**–**e**) The magnetic field imaging results at pitch angles of 0°, 1°, 2°, 3°, and 4°.

**Figure 8 sensors-24-02613-f008:**
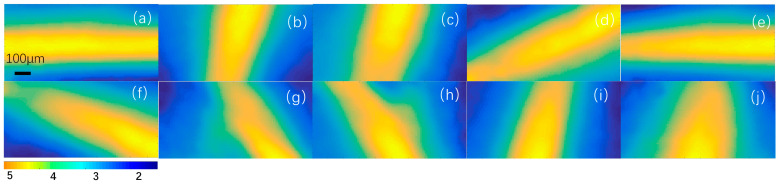
Comprehensive testing of yaw angle, pitch angle, and roll angle. (**a**–**j**) The magnetic field imaging results at different yaw angles, pitch angles, and roll angles.

**Figure 9 sensors-24-02613-f009:**
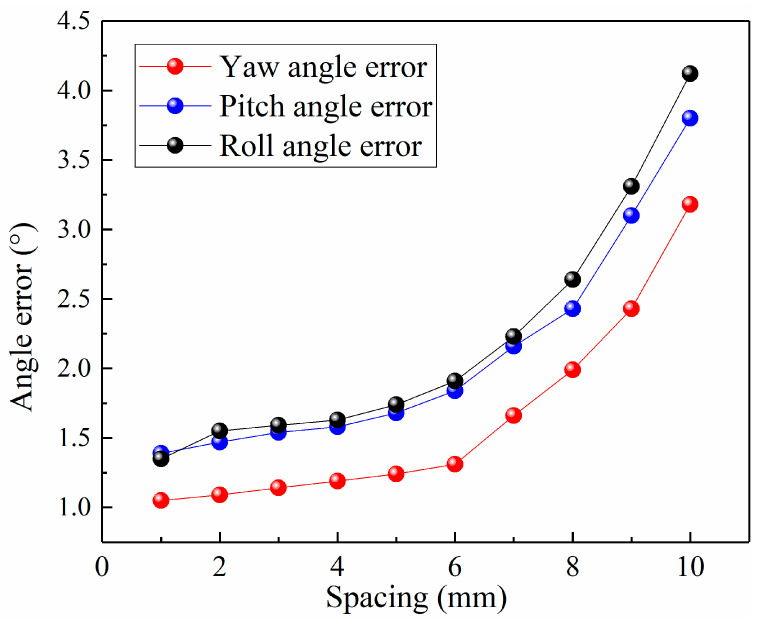
The schematic diagram of the experimental device.

**Table 1 sensors-24-02613-t001:** The relationship between spacing and angular resolution.

Spacing (mm)	Yaw Angle *α* (°)	Pitch Angle *φ* (°)	Roll Angle *β* (°)
1	1.05	1.39	1.35
2	1.09	1.47	1.55
3	1.14	1.54	1.59
4	1.19	1.58	1.63
5	1.24	1.68	1.74
6	1.31	1.84	1.91
7	1.66	2.16	2.23
8	1.99	2.43	2.64
9	2.43	3.1	3.31
10	3.18	3.8	4.12

## Data Availability

Data are contained within the article.

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
