# Peer review of "Measurements of Spatial Angles Using Diamond Nitrogen–Vacancy Center Optical Detection Magnetic Resonance"

_sensors, 2024, doi:10.3390/s24082613_

Round 1
Reviewer 1 Report
Comments and Suggestions for Authors
The manuscript “Measurements of Spatial Angles Using Diamond NV Color Center Optical Detection Magnetic Resonance” is devoted to the development of a technique for determining the angular coordinates of small magnetic field sources. Although the experimental part of the work as a whole makes a good impression, the manuscript in its present form is replete with shortcomings that need to be corrected.
1. Firstly, the introduction raises questions. The authors compare their method with existing ones, such as “total stations [5], laser trackers [6,7], iGPS [8,9], and MScMS [10-12]”. Considering that the authors propose a method for determining the positions of small (sub-centimeter) sources of magnetic fields, comparison with total stations, laser trackers and iGPS does not seem entirely appropriate. The method for localizing small sources of magnetic disturbance must be compared with similar magnetic methods.
2. Further, the authors write: “Magnetic spatial angle measurements have strong anti-interference capabilities against external disturbances, especially being able to work stably and reliably even when obscured.” This statement is incorrect in the vast majority of cases, since the result of magnetic measurements (unlike, for example, the same laser trackers) is destructively affected by spatiotemporal variations of both the earth’s magnetic field and industrial magnetic interference. Eliminating these interferences is a complex technical task and requires either the use of gradiometric methods or the creation of complex and expensive magnetic shields or magnetically insulated rooms. If we add to this the property of the magnetic field of a dipole to decrease in proportion to the cube of the distance, we will see that variations in the Earth's field impose strict restrictions on the detection of fields from small objects.
3. Section 2.1. “Magnetic Field Detection Based on ODMR” (almost three pages) contains exclusively well-known facts (mostly given without references). In general, it is not entirely clear what exactly is new in the method proposed by the authors. This should be clarified by making an explicit comparison with the work of predecessors. At the same time, the proposed method is not described fully enough, in particular, the authors do not mention how exactly they measure the frequencies of the eight resonances shown in Fig.1.
4. The authors use a diamond plate measuring 5x5 mm2 as a sensor. If this is not the limit of the size of a synthetic diamond, it is very close to the limit. Thus, the sensor size in this method is limited; it is also limited by the laser power (0.5 W/mm2, that is, 12.5 W for these plate sizes!). Accordingly, the distance to the object at which the angular resolution specified by the authors is achieved is also limited.
Indicating the angular resolution (including in the annotation) does not make sense without indicating the parameters of the magnetic dipole and the distance to it. The manuscript should be supplemented with missing information. Some of this information is given in the text, for example “a current of 0.5A is passed through the chip to image the electromagnetic field generated when powered” (quite a lot of current for a chip!). As far as I can tell, neither the chip size nor the distance to it is specified.
The authors should add this information, as well as a table of angular resolution values depending on the parameters of the object and the distance to it.
It would also be very desirable to provide the signal-to-noise ratios achieved in the experiment, as well as their comparison with the maximum achievable (for example, limited by photon shot noise) values.
5. It should also be noted how the measurement accuracy is affected by the thickness of the diamond plate, as well as the spatial inhomogeneity of the pump light, microwave and magnetic fields.
6. Authors should also indicate the required measurement time.
7. It should be explained what determines the choice of the displacement field (3 mT), and how this field is related to the parameters of the measured object.
8. The numbering of the list of references is confused (see, for example, “Since the combination of diamond NV centers and magnetic resonance technology in 2008 [18]...”.
9. Fig.5-7 lacks informative captions.
All this makes further analysis of the manuscript impossible. I believe that the manuscript cannot be printed in its present form, and must be radically revised.
Reviewer 2 Report
Comments and Suggestions for Authors
The manuscript "Measurements of Spatial Angles Using Diamond NV Color Center Optical Detection Magnetic Resonance" presents a spatial angle measurement system that capitalizes on the precision of ensemble diamond NV center ODMR technology. This approach addresses the limitations of existing measurement methods by offering a non-contact, interference-resistant solution, significantly enhancing accuracy and applicability in various fields. Through the performed experimentation, the system demonstrated remarkable measurement accuracy, with errors reduced to 1° for yaw angles and 1.5° for pitch and roll angles. This research not only validates the system's superior performance against established benchmarks but also introduces a transformative tool for advanced, non-invasive spatial measurements, promising widespread implications for industries ranging from aerospace engineering to biomedical research. I recommend a publication of this work after addressing the following points.
Comments and suggestions:
1. Please quantify and highlight the key finding of this work in the abstract, introduction, and conclusion.
2. The paper should discuss potential barriers to adoption, including the complexity of the technology, the expertise required to operate it, and the associated costs. Suggestions for making the technology more accessible to a broader audience would be valuable.
3. A more comprehensive comparison with traditional spatial measurement technologies, including aspects such as cost, ease of use, and applicability in different environments, would provide readers with a better understanding of the new system's relative advantages and limitations.
4. Please perform a comprehensive reading of the manuscript to avoid typos.
Comments on the Quality of English LanguageMinor editing
Reviewer 3 Report
Comments and Suggestions for Authors
The manuscript describes results of spatial angles measurements using diamond NV centers optically Detected Magnetic Resonance. They collected fluorescence signals from a 5×5 mm2 diamond with concentration of NV centers of about 1.5 ppm and transforming them into ODMR data to calculating magnetic field information. With the information, they derive the relationship between angle and magnetic field intensity. I think the purposes for exploring the spatial angles measurement applications using diamond NV centers are significant. However, there are many problems in the manuscript. These include, but not limited follows:
1. In background part, the authors do not mention literature on the spatial angle measurement. They should add more about this aspect or highlight the initiative of their work. Meanwhile, the introduction is suggested to be re-organized and modified in deep.
2. The English of the full text needs to be carefully revised. For example, the author uses “NV color center” and “NV center”, can the authors unify them? The full name of ODMR should be optically detected magnetic resonance.
3. The numbering and description of the figures are problematic and need to be revised. For example, the 8th picture should be Figure 8 not 2. The descriptions of Figure 5, 6 and 7 are the same and have nothing to do with the content of the pictures. Line 168, Where is the fig. 2(d). I also suggest that Figure 1(a) and 1(b) be reversed since the Figure 1(b) was first mentioned up.
4. With what experimental conditions, the spectrum in Fig. 1(c) was obtained, can the authors give the details about conditions.
5. Line 134, what is the physical meaning of m for?
6. I suggest that the authors provide some relevant literature to cite the reasonableness of the derivation of the formula.
7. Please add more details in Figure 4 to describe the position of the diamond and chip.
8. Please add more details such as scale bars, angle description, in Figure 5, 6 and 7 to have a clear understanding for these results.
9. The x-coordinates in Figure 8 do not indicate physical quantities.
10. Can the authors provide an assessment of yaw angle error in other literature?
In my opinion, the research in this manuscript has not yet met the acceptance criteria of the Sensors journal and other journals can be considered after major revision.
Comments on the Quality of English LanguageThe English is only acceptable, I advice authors to ask an expert in English for a careful revision of the MS the prior to the resubmission
Round 2
Reviewer 1 Report
Comments and Suggestions for Authors
First of all, I must admit that the authors did a great job and responded to all the reviewer's comments, providing detailed answers and making appropriate changes to the manuscript. I am grateful to the authors for the kind words they said to the reviewer, but I am afraid that the reviewer does not deserve such excellent epithets.
In general, I am satisfied with both the answers and the corrections in the text. When reading the latest version of the manuscript, I had several questions and comments:
1. L149 “In the absence of an external magnetic field, the ms = ±1 spin states are degenerate” - in the general case this is not so. Due to the transverse zero-field splitting, anti-crossing of levels occurs, preventing degeneration.
2. L256 “The diamond NV center is imaged on the focal plane of the CCD, with a resolution of 1920 × 1080 pixels and a size of 5 μm² - what does 5 μm² refer to here - the area of an individual pixel? Please explain.
3. Lines 301-305 basically repeat lines 257-261.
4. It might be useful to depict in more detail in Fig.4 (in the corresponding inset) part of the scheme including the diamond, focusing systems, CCD and the object under study.
5. It might also be useful, in addition to the processed images (Fig. 5-7), to show some raw images directly from the camera.
I believe that after making corrections on points 1-3, the manuscript can be published; Points 4-5 remain at the discretion of the authors.
Reviewer 3 Report
Comments and Suggestions for Authors
I think the revised version meets the stantard of the SENSORS journal。There still some questions exist. For example, the authors did not explicitly quantitatively compare their results with other work. I hope these problems will be explained in their future studies
Comments on the Quality of English LanguageThe English is acceptable, I advice authors to ask an expert in English for a careful revision of the MS the prior to the resubmission
